# Study on the Applicable Room Size Dimension of Stratum Ventilation for Heating Based on Multi-Criteria Analytic Hierarchy Process-Entropy Weight Model

Yanhui Mao [1],* , Honglei Xie [2], Xinlu Zhang [1,3], Fumin Hou [1,4] and Miantong Wang [4,5]

[1] Department of Building Environment and Energy Application Engineering, Ningbo University of Technology, Ningbo 315211, China
[2] Ningbo University Architectural Design and Research Institute Co., Ltd., Ningbo 315211, China
[3] School of Engineering and Architecture, Zhejiang Sci-Tech University, Hangzhou 310018, China
[4] College of Mechanical Engineering, Tianjin University of Commerce, Tianjin 300134, China
[5] Tianjin Ouke Environmental Equipment Co., Ltd., Tianjin 301721, China
* Correspondence: maoyanhui@nbut.edu.cn

**Abstract:** With the implementation of clean heating and the outbreak of COVID-19, stratum ventilation with both energy-saving and healthy indoor environments has become a research hotspot. Room size dimension is one of the critical factors affecting the air distribution, thermal comfort, and ventilation performance of space heating, which is still a research blank at present. This study determined the applicable room size dimension of stratum ventilation for space heating by using a multi-criteria analytic hierarchy process-entropy weight (AHP-EW) model. A computational fluid dynamics (CFD) simulation verified by experiments was conducted. To investigate the ventilation performance of different room sizes in energy utilization and thermal comfort, airflow distribution, ventilation efficiency ($E_t$), dimensionless temperature, effective ventilation temperature (EDT), air distribution performance index (ADPI), and predicted mean vote (PMV) were calculated. The multi-criteria AHP-EW method is used to evaluate every case comprehensively. The results show that the maximum room size obtained by multi-criteria APH-EW is 6 m, and considering the single criteria, the suitable height for stratum ventilation for heating is below 5.7 m. The data obtained in this paper can be used as a reference for further study on the application of stratum ventilation and heating in the future.

**Keywords:** applicable room size dimension; muti-criteria evaluation; stratum ventilation; space heating; AHP-EW model; operation optimization

## 1. Introduction

The outbreak of COVID-19 has brought significant challenges to people's economies and lives [1]. In response to the epidemic, various measures have been taken to prevent the spread of the virus. Relevant studies have shown that in addition to droplet and contact transmission, aerosol transmission is also the primary way of COVID-19 transmission, especially in closed spaces [2]. Because aerosol is greatly affected by airflow, ventilation is considered to be an effective way to reduce air transmission [3]. Reasonable air distribution is widely used to create satisfactory and healthy indoor thermal environments [4–6]. Through effective ventilation, the concentration of aerosol-carrying virus is diluted, and the diffusion path is blocked [7]. However, incorrect air distribution will not only disturb the airflow organization but also increase the risk of infection [8]. In addition, the recycling in the air-conditioning system is canceled, and the application of fresh air is expanded, which leads to an increase in energy consumption and makes people start to look for healthy and low-carbon indoor environment creation methods [9].

Mixing ventilation, as the conventional air distribution, aims to condition the entire space uniformly [10,11]. This usually causes a waste of energy and cannot satisfy people's

comfort. At the same time, due to the inability to eliminate indoor polluted air, the range and concentration of indoor infectious particles are increased [12]. For a better thermal environment and energy efficiency, air distributions with a non-uniform thermal environment are preferred to condition only the occupied zone, e.g., personalized ventilation [13] and stratum ventilation [14]. Stratum ventilation is a novel air distribution, proposed for application to small-to-medium-sized spaces [15]. The air supply outlet of the stratum ventilation is located at 1.1 m on the side wall. By delivering fresh air directly to the breathing zone and occupied zone, the energy efficiency and air quality in indoor environments can be ensured. Compared with mixing ventilation and displacement ventilation, stratum ventilation can save energy annually by at least 44.37% and 25.61% through constructing a non-uniform thermal environment [16]. From the aspect of thermal comfort, the study by Cheng et al., showed that the distribution of temperature and air velocity in the personal zone was quite uniform, and the air diffuser performance index (ADPI) could reach more than 80% [17]. The subjective experiment voting results showed that the highest acceptance percentage of stratum ventilation was also higher than mixed ventilation and displacement ventilation, and thermal comfort could be achieved at higher air-supply temperature conditions [18]. The advantages of stratum ventilation in building a healthy and energy-saving indoor environment are reflected. To achieve a good operation performance of stratum ventilation, existing studies have mostly focused on feasibility studies of ventilation parameters by subjective and objective experiments. It was reported that increasing the supply air temperature from 18.5 to 21.5 °C can cause the slightly cool thermal environment to become slightly warm and to reduce the draft rate from 37% to 2% with a supply airflow rate of 10 air changes per hour (ACH) [19]. Furthermore, the supply airflow rate and the supply air temperature were revealed to interactively affect the energy efficiency by Huan et al. in their experiment [20]. Cheng et al. [21] and Zhang et al. [22] proposed optimization methods for the room air temperature and the fresh outdoor air ratio to maximize energy efficiency while achieving thermal comfort and good air quality. In addition, Zhang et al. [22] and Kong et al. [23] also employed multi-criteria decision-making technology and the Marquardt method to comprehensively optimize the supply vane angle, supply airflow rate, and supply air temperature, by considering thermal comfort, air quality, and energy efficiency. Meanwhile, Zhao et al. [24] pointed out that the supply air temperature and grille angle obviously affected thermal comfort and energy efficiency. PPD achieved the minimum value with the supply air temperature being 20 °C and the maximum with the supply air angle being 90 degrees. Tian et al. [25] found that stratum ventilation may offer a feasible solution to elevated indoor temperatures in summer by adjusting the supply air temperature. Cheng et al. [19] proposed that a thermally comfortable environment was easier to realize when the supply air temperature was higher than 27 °C due to the feeling of "cool head and warm feet".

Previous studies indicated that stratum ventilation was usually applied to small-to-medium-sized spaces for the reason that its supply air outlets were close to the occupied zones, and thus, it was easy to cause the draft sensation. Moreover, the proper operation of stratum ventilation for heating is more complicated than that for cooling. Hence, the findings on stratum ventilation for cooling are not applicable to stratum ventilation for heating; the optimization of the stratum ventilation for heating is required to improve operational performance. Some related studies confirmed the energy-saving potential and the advantage of providing a comfortable environment for stratum ventilation used for heating [23,26] and conducted a series of optimization studies on the supply air parameters [27]. However, little attention was paid to the applicable room size dimension of stratum ventilation for space heating, which obviously impacts its application for heating. Under the heating condition, stratum ventilation supply airflow movement is dominated by inertial force and thermal buoyancy. The room size dimension can change the jet path and affect the formation of the eddy zone. Meanwhile, to ensure the accessibility of the warm air flow in the space, higher supply-air velocity will be employed as the room size dimension increases, which may cause energy waste and personal discomfort. However,

with the room size dimension as one of the critical factors affecting air distribution, thermal comfort and performance of stratum ventilation used for space heating is still a research gap. Therefore, it is necessary to clarify the applicable room size dimension of stratum ventilation for space heating to improve the operational performance further and enhance suitability.

Determining the applicable room size dimension of stratum ventilation for heating is a multi-criteria decision-making problem. At different room size dimensions, the conflict between energy efficiency and thermal comfort will be an urgent issue. The analytic hierarchy process (AHP) and entropy weight (EW), as common multi-criteria decision making techniques, are often used to evaluate the rationality of indoor environments [27,28]. For example, Ren et al. used AHP to optimize the carriage ventilation system by comprehensively balancing the conflict between infection risk, thermal comfort, and energy consumption in subway carriages [28]. Zhang et al. used EW to optimize the heating operation of stratum ventilation and obtained the optimal air supply parameters during outdoor weather changes [27]. However, AHP and EW can only consider subjective or objective preferences of a particular aspect when used alone and are susceptible to being influenced by expert intent or data extremes. The AHP-EW method with both subjectivity and objectivity can compensate for these deficiencies and thoroughly weigh the conflict between energy efficiency and thermal comfort.

Hence, as per the aim of this study, the applicable room size dimension for stratum ventilation for space heating was optimized based on the multi-criteria AHP-EW method. The flow chart for the current study is schematically shown in Figure 1. Experimentally validated Computational Fluid Dynamics (CFD) models with the room size dimension changing from 3 m to 10 m were employed in this study. First, single criteria were used to assess the environment of different room sizes. Airflow distribution, ventilation effectiveness ($E_t$), and dimensionless temperature were used to represent energy use efficiency; effective draft temperature (EDT), air distribution performance index (ADPI), and variations of predicted mean vote (PMV) were used to evaluate thermal comfort. Secondly, AHP and EW were combined to comprehensively judge the performance in different room sizes. Finally, a multi-criteria AHP-EW evaluation system was established. Based on the single criteria and AHP-EW results, the applicable room size dimension of stratum ventilation for space heating was obtained. The main contributions of this study are (1) clarifying the applicable room size dimension of stratum ventilation for space heating and (2) establishing a multi-criteria AHP-EW evaluation system to simultaneously consider subjective factors, objective factors, and single criteria. The data obtained in this paper can be used as a reference for future in-depth studies on the heating application of stratum ventilation.

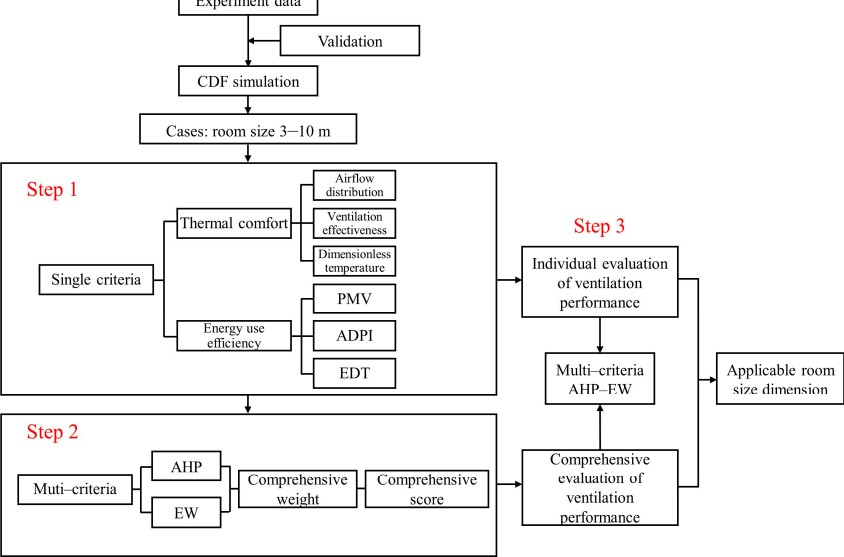

**Figure 1.** Study flow chart.

## 2. Methodology

### 2.1. Experiment

The experiment was conducted in a typical office test chamber. The test chamber's length, width and height are 6 m, 4 m, and 3.5 m, respectively. This test chamber can simulate the air distribution of stratum ventilation. The experiment was conducted at an outdoor ambient temperature of −3 to 10 °C, which are common weather conditions for winter heating.

The fresh air was supplied by two rectangular double grid diffusers (0.3 m × 0.3 m) located at 1.2 m of the side wall and sent out by the louver (0.3 m × 0.3 m) at the ceiling. The rotational speed of the fan was adjustable, to deliver different volumes of supply air. A dry bulb temperature sensor with an accuracy of 0.1 °C and a velocity sensor with an accuracy of 2% were used to measure the air supply temperature and velocity. To realistically recreate an office scenario, six cuboids (0.4 m × 0.25 m× 1.2 m) were placed in the test chamber to simulate sedentary occupants. A 0–100 W bulb was placed inside the cuboid to simulate the heat dissipation of the human body. This method is considered effective for indoor thermal environment experiments [20]. In addition, two 75 W lamps were placed at the ceiling. The experiment was conducted in winter, and thermocouples were set on the surface of the enclosure structure to monitor the boundary conditions of the test chamber.

Ambient temperatures and velocities in the laboratory were monitored and recorded. Five measuring points were set, and the measuring points were arranged in a line with a height of 1.1 m in the test chamber. Figure 2 shows the specific location of the measuring points. The temperature and velocity of measuring points were measured by a Testo 480 hot-line anemometer and T-thermocouple. The specific precision and measuring range of the experimental instrument are shown in Table 1. In addition, as a data collector, Agilent was used to convert thermal signals to electrical signals and save temperature data by connecting with the T-thermocouple. The physical view of the measurement instrument is shown in Figure 3.

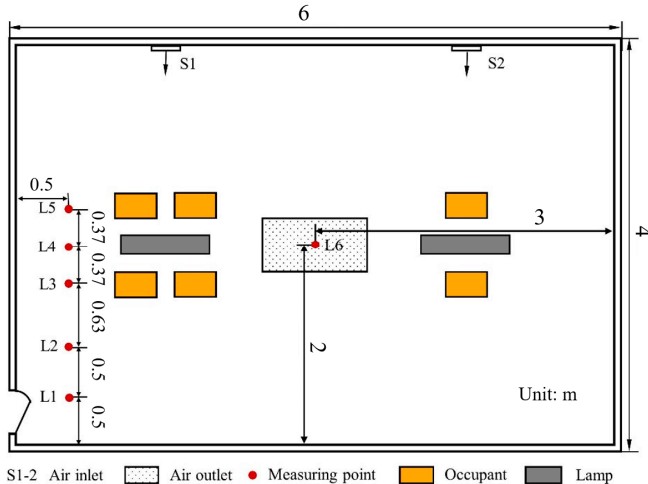

**Figure 2.** Layout of the measurement points.

**Table 1.** Details of the measurement instruments.

| Measured Parameter | Measurement Instrument | Measurement Range | Measurement Accuracy |
|---|---|---|---|
| Air temperature | T-thermocouple | −200~260 °C | ±0.1 °C |
| Air velocity | Testo480 hot-line anemometer | 0~20 m/s | ±0.05 m/s |

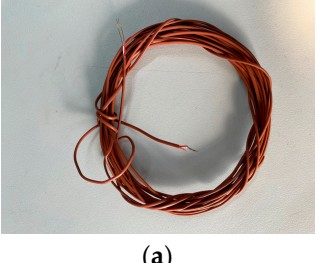
(**a**)

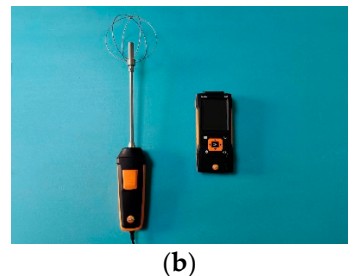
(**b**)

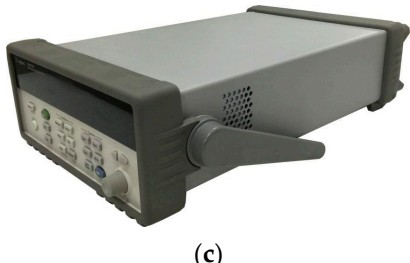
(**c**)

**Figure 3.** Physical view of the measurement instrument: (**a**) T-thermocouple; (**b**) Testo 480 hot-line anemometer; (**c**) Agilent.

### 2.2. Computational Fluid Dynamics (CFD) Simulation

Experimental measurements and CFD simulations are the two main research methods employed in this study to investigate the various indoor air distributions [29,30]. Experimental measurement can provide realistic data, but this method is expensive and time-consuming. Thus, CFD simulation was added to this study. With the rapid development of computational capabilities, CFD simulations of lower cost are widely used to predict indoor airflow, indoor air temperature, thermal comfort, indoor air quality, etc. [31]. Experimental measurements can be collected to validate the reliability of CFD simulations. The experimentally validated CFD simulations are used to produce data for the development and validation of response surface models. This is a common practice in the field of the built environment [32].

#### 2.2.1. Boundary Conditions

To ensure the high quality of the CFD simulation of an indoor thermal environment, verification and validation followed a standard procedure used in this study [33]. Airpak 3.0.16 was used to construct the physical model, generate mesh, and perform the simulation.

To obtain reasonable accuracy and relatively high computational efficiency, the standard k-ε model was adopted in this study [34]. Compared with the RNG k-ε model and the SST k-ω model, the standard k-ε model is more suitable for simulating air temperature and airflow under ventilation conditions. The airflows in the room were assumed to be steady-state, incompressible, and turbulent. The radiant heat transfer between surfaces was considered in the discrete ordinates radiation model. Additionally, the buoyancy effect was computed using the Boussinesq mode. The airflow velocity of outlets in the model was assumed to be uniform, and the inlets were specified as the Neumann boundary condition [23,35]. The standard wall function was employed to model the turbulent flow in the near-wall region [36]. Moreover, to maintain the indoor supply and demand load balance, the size of the supply air outlets was changed [37] with the room size dimension. The occupants and lights were defined as a constant heat flux solid boundary condition with a power of 100 W and 75 W, respectively. The detailed boundary conditions are summarized in Table 2.

**Table 2.** Boundary conditions of CFD simulations.

| Boundary | Boundary Condition | Numerical Parameter |
|---|---|---|
| Inlet | Uniform velocity inlet | 27 °C, 1.94 m/s |
| Outlet | Free outflow | |
| Occupant | Constant heat flux solid boundary | 100 W |
| Light | Constant heat flux solid boundary | 75 W |
| Outer wall | Constant wall temperature | −3.2 °C |
| Internal wall | Constant wall temperature | 12.5 °C |
| Left/right wall | Constant wall temperature | 16.4 °C |
| Ceiling | Constant wall temperature | 14.5 °C |
| Floor | Constant wall temperature | 14.5 °C |

2.2.2. Mathematical Models

Navier-Stokes equations were considered as the control equations of the indoor airflow in the numerical simulation; the mass continuity equation, momentum equation and energy equation are shown below [38].

The mass continuity equation denotes that the mass flowing into the control body micro-element per unit time is equal to the mass of the control body micro-element increase. The equation is as follows:

$$\frac{\partial \rho}{\partial t} + \nabla \cdot (\rho U) = 0 \tag{1}$$

where $\rho$ is the density of the fluid, kg/m$^3$; $t$ is the time, s; $U$ is the fluid flow velocity, m/s.

The momentum equation is the sum of all external forces on the micro-element of the control body equal to the increase in momentum of the micro-element of the control body. The equation is as follows:

$$\frac{\partial (\rho U)}{\partial t} + \nabla \cdot (\rho U \otimes U) = -\nabla p + \nabla \cdot \tau + S_M \tag{2}$$

where Stress tensor $\boldsymbol{\tau}$ is related to strain rate, $\tau = \mu \left( \nabla U + (\nabla U)^T - \frac{2}{3} \delta \nabla \cdot U \right)$; $T$ is thermodynamic temperature, °C; $\delta$ is identity matrix; $p$ is thermodynamic compressive strength, Pa; $\mu$ is dynamic viscosity, N·s/m$^2$; $S_M$ is fluid momentum source.

The energy equation states that the total energy change per unit volume of fluid is only equal to the sum of the inflow or outflow of energy of the fluid. The equation is as follows:

$$\frac{\partial \rho h_{tot}}{\partial t} - \frac{\partial \rho}{\partial t} + \nabla \cdot (\rho U h_{tot}) = \nabla \cdot (\lambda \nabla T) + \nabla \cdot (U \cdot \tau) + U \cdot S_M + S_E \tag{3}$$

where $h$ is fluid static enthalpy, J/kg; $\lambda$ is fluid thermal conductivity, w/(m·k); $S_E$ is fluid energy source.

2.2.3. Grid-Independent Tests

The previous sections prepared the physical and mathematical conditions for the CFD simulations to be carried out. To further ensure the accuracy and efficiency of the simulation, grid-independence verification was required before the simulation. The grid numbers of the three schemes were 914,568 (coarse grid), 1,803,536 (moderate grid), and 2,772,928 (fine grid), respectively. Grids near inlets, outlets, and human bodies were encrypted for higher accuracy. To reduce the impact of occupants and walls, points with different heights at L6 in the center of the room were selected as comparison points. The specific location of L6 is shown in Figure 2.

As shown in Table 3, the verification results indicate that the temperature and velocity under the coarse grid are quite different from those of the moderate grid and the fine grid; the maximum relative temperature error is 2.19%. However, the maximum relative error under the moderate grid and the fine grid is only 0.51%, which is close to 0. When the calculation accuracy is similar, considering the economic cost of calculation, the moderate grid was adopted in this study, and its grid quality is higher than 0.9.

**Table 3.** Grid independence verification.

| | Grid Numbers | 0.5 m Height (°C) | 1 m Height (°C) | 1.5 m Height (°C) | 2 m Height (°C) | 2.5 m Height (°C) | Maximum Relative Error (%) |
|---|---|---|---|---|---|---|---|
| 1 | 914,568 | 18.3 | 18.6 | 18.9 | 19.1 | 19.3 | 2.19 |
| 2 | 1,803,536 | 17.9 | 18.4 | 18.8 | 19.0 | 19.0 | 0.51 |
| 3 | 2,772,928 | 18.0 | 18.5 | 18.8 | 19.0 | 19.0 | —— |

2.2.4. Simulation Validity Verification Method

The experimental and simulated air supply parameters were set to be the same: the air supply temperature was 26 °C, the speed was 1.94 m/s, and the air supply angle is horizontal. At the same time, the boundaries remained the same. Relative error and the mean absolute percentage error (MAPE) can quantitatively evaluate the model's accuracy. To evaluate the model's accuracy quantitatively, relative error and the mean absolute percentage error (MAPE) [39] were used to evaluate the results. The relative error is defined as follows:

$$E = \frac{\Delta}{L} \times 100\% \tag{4}$$

where $E$ is the actual relative error, which is given in percentage. The $\Delta$ is the absolute error; $L$ is true.

The mean absolute percentage error is defined as Equation (14):

$$\text{MAPE} = \frac{1}{P} \sum_{i=1}^{p} \frac{|x_i - y_i|}{|x_i|} \times 100\% \tag{5}$$

where $x_i$ is the experimental result; $y_i$ is the numerical simulation result; $P$ is the number of experimental measuring points. When the experimental results are completely consistent with the numerical simulation results, the average absolute percentage error should be 0, and the consistency index should be 1.

*2.3. Simulated Cases*

The room width was treated as the characteristic length to obtain the applicable room size dimension of stratum ventilation. The room width of studied cases covers commonly encountered values in life; it was set at 8 levels (3 m, 4 m, 5 m, 6 m, 7 m, 8 m, 9 m, and 10 m). The supply air speed and temperature were set as 1.94 m/s at 27 °C. The air supply angle was horizontal and kept the same heat flux under different conditions.

*2.4. Ventilation Performance Evaluation Method*

First, single criteria were used to evaluate each case's energy utilization efficiency and thermal comfort separately. Second, the multi-criteria analytic hierarchy process-entropy weight method was adopted to calculate the score of each case synthetically. Based on the comprehensive score, the applicable room size dimension of stratum ventilation for space heating was obtained. Moreover, single index evaluation results were also added to get the final result.

2.4.1. Single Criteria

To evaluate the optimum room size for stratum ventilation, a hierarchical model was adopted here. The details are shown in the following Figure 4. Three indicators representing energy use efficiency and three indicators for thermal comfort were selected to analyze the performance of different cases, respectively. The index selection and calculation methods are further described as follows.

1.    Energy use efficiency

Energy use efficiency reflects the energy saving of ventilation mode. Because of the special airflow mode of stratum ventilation, the change in room size dimension will lead to a change in energy consumption. In this section, the airflow path was reflected through the velocity cloud atlas directly, and the stay of fresh air was expressed by the mean age of air first. Second, taking ventilation efficiency and dimensionless temperature as indices, the residence and stratification of heat in the occupied zones of different sizes of rooms were judged, and the size of the stratum ventilation room with the highest energy utilization efficiency was finally obtained.

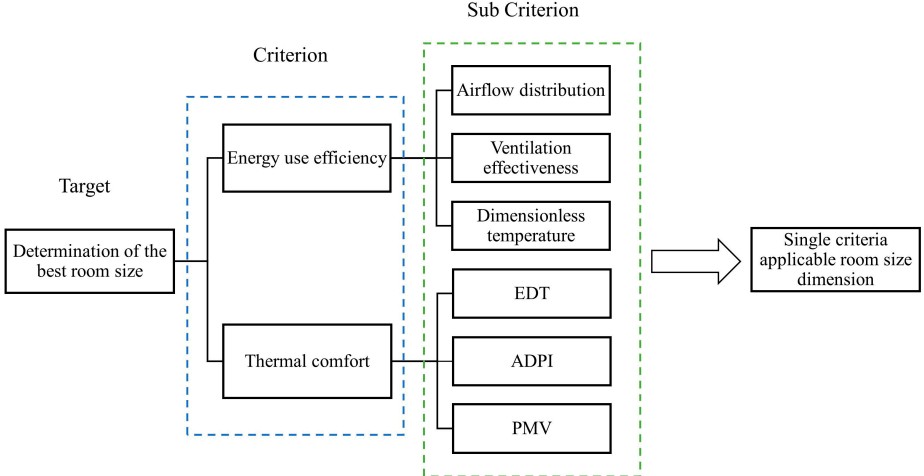

**Figure 4.** Hierarchical model for the evaluation of suitable room size.

Air flow distribution refers to fresh air and the exchange with indoor air. The rise of flowing air due to thermal buoyancy was obtained by air velocity profile, and the stay of fresh air in the occupied zone was calculated by the mean age of air (MAA). Combining the above two indicators, the energy utilization efficiency is reflected.

The MAA value reflects the flow characteristics of the air supply, so it can be used to evaluate air supply distribution [40]. The mean age of air $\tau_p$ is given as follows [41]:

$$\tau_p = \frac{\int_{t_0}^{\infty} C_t dt}{C_{t-t_0}} \tag{6}$$

where $\tau_p$ is the local mean age of air, s; $t$ is the time, s; $C_{t-t0}$ is the initial tracer gas concentration at time $t - t_0$, ppm.

Ventilation effectiveness ($E_t$) [42]:

$$E_t = (T_s - T_e)/(T_s - T_o) \tag{7}$$

where $E_t$ is ventilation efficiency; $T_s$ is the supply air temperature, °C; $T_e$ is the return air temperature, °C; $T_o$ is the average temperature in the occupied zone, °C. Ventilation efficiency is related to system energy utilization efficiency. Higher ventilation efficiency means that the hot-air flow sent to the room can stay in the occupied zone more effectively.

Dimensionless temperature [43]:

$$\varphi_H = \frac{t_e - t_s}{t_H - t_s} \tag{8}$$

where $\varphi_H$ is dimensionless temperature; $t_e$ is return air temperature, °C; $t_H$ is plane average temperature, °C; $t_s$ is air supply temperature, °C. Dimensionless temperature can be used to represent the thermal stratification in the room. The closer the dimensionless temperature is to 1, the less obvious the thermal stratification phenomenon, which means that the indoor air is thoroughly mixed.

2.  Thermal comfort

Because of the special airflow mode of stratum ventilation, indoor comfort can obviously be affected by room size. In this section, EDT was shown to evaluate air velocity and temperature distribution uniformity. ADPI was calculated to evaluate the satisfaction rate of indoor draft sensation and air velocity. Combined with relative humidity, human activities, and clothing thermal resistance, PMV was employed to evaluate the overall thermal comfort.

Effective draft temperature (EDT) [44,45]:

$$\text{EDT} = T_x - T_r - (v_x - 1.1) \tag{9}$$

where $T_x$ is the temperature of a certain point in space, °C; $T_r$ is the average indoor temperature, °C; $v_x$ is the air velocity at a certain point in indoor space, m/s. To describe the uniformity of the synergistic effect of the velocity and temperature for stratum ventilation, Lin recently proposed a new formula of the EDT for stratum ventilation, which is found to be reliable and straightforward in the evaluation of the performance in thermal comfort.

Air distribution performance index (ADPI) [10]:

$$\text{ADPI} = \frac{N_s(-1.2\text{K} < \text{EDT} < 1.2\text{K}, 0 < v < 0.8\text{m/s})}{N} \times 100\% \tag{10}$$

where $N$ is the total number of occupied zones; $N_s$ is the total number of effective blowing temperatures and air velocities in the comfortable range. ADPI is directly proportional to the comfort of the indoor thermal environment, and when ADPI is 100%, the indoor thermal environment is the most comfortable

Predicted mean vote (PMV) [45,46] is an evaluation index of human thermal sensation. A PMV of 0.1–1.8 m was selected for thermal comfort assessment. Among them, the standard clothing thermal resistance in winter is 1.5 clo.

### 2.4.2. Multi-Criteria Analytic Hierarchy Process-Entropy Weight Method (AHP-EW Model)

The above subsection describes the method of single criterion for obtaining the optimal room size. However, due to a large number of indicators and complex data, making a direct determination of the best room size based on a single criterion is problematic. It is of great significance for getting the comprehensive weight of each index by the AHP-EW method to make sure the calculation is reasonable [47–50]. Therefore, by combining the single criteria with the AHP-EW model, this study proposed the multi-criteria AHP-EW model for ventilation performance evaluation. The process of determining the applicable room size dimension using this method is shown in Figure 5. Steps 1–4 describe the calculation procedure in detail.

Step 1: determining the score. According to the simulation results, the score of each room size dimension was determined. The threshold method was adopted to make the data standard and dimensionless. The data standardization process is as follows [51]:

Positive indicator:

$$x_{ij}' = \frac{x_{ij} - \min\{x_{ij}, \cdots, x_{nj}\}}{\max\{x_{1j}, \cdots, x_{nj}\} - \min\{x_{ij}, \cdots, x_{nj}\}} k + q \tag{11}$$

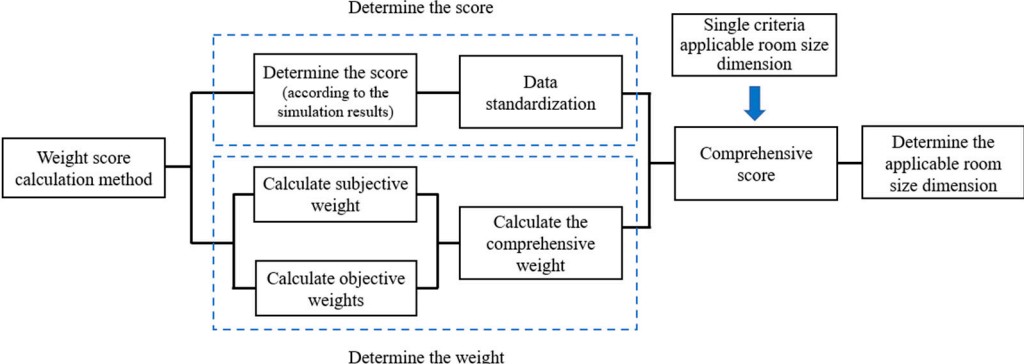

**Figure 5.** Process to determine the applicable room size dimension.

Negative index:

$$x_{ij}' = \frac{\max\{x_{1j}, \cdots, x_{nj}\} - x_{ij}}{\max\{x_{ij}, \cdots, x_{nj}\} - \min\{x_{1j}, \cdots, x_{nj}\}} \cdot k + q \tag{12}$$

Where, $x_{ij}'$ is the standardized score of the $j_{th}$ room size dimension in the $i_{th}$ index; $x_{ij}$ is the initial score of the $j_{th}$ room size dimension in the $i_{th}$ index; $k$ and $q$ are coefficients.

Step 2: determining the comprehensive weights by the AHP-EW method [52,53]:

$$\beta_i = \frac{\alpha_i W_i}{\sum\limits_{i=1}^{n} \alpha_i W_i} \tag{13}$$

where $\alpha_i$ is the weight calculated by AHP for the $i_{th}$ index; $W_i$ is the weight calculated by the EW method for the $i_{th}$ index; $\beta_i$ is the comprehensive weight combining AHP and entropy weight for the $i_{th}$ index.

Step 3: calculating the comprehensive score. The calculation method of comprehensive score is described as follows:

$$S_j = \sum\limits_{i=1}^{n} \beta_i x_{ij} \tag{14}$$

where $S_j$ is the comprehensive weight score for the $j_{th}$ size room; $\beta_i$ is the comprehensive weight for the $i_{th}$ index; $x_{ij}$ is the standard scores for the $i_{th}$ index of the $j_{th}$ size room.

Step 4: obtaining room size dimension. By adopting the AHP-EW score and single criteria, the largest room size under each index was found and the applicable room size dimension is obtained.

## 3. Results and Discussion

### 3.1. Validation of CFD Simulations

Figure 6 shows the comparison results between simulated data and experimental data of measuring points 1–5. The results show that the experimental and simulated temperature trends are the same, and the numerical values are in good agreement. Compared with the simulated values, the relative error of the experimental values is less than 5%, and the average absolute percentage error is 0.49%, close to 0. Therefore, it can be shown that the temperature results of the numerical simulation are basically consistent with the experimental results. It shows that the numerical model has high accuracy and can be applied to the simulation calculation of air distribution in stratum ventilation.

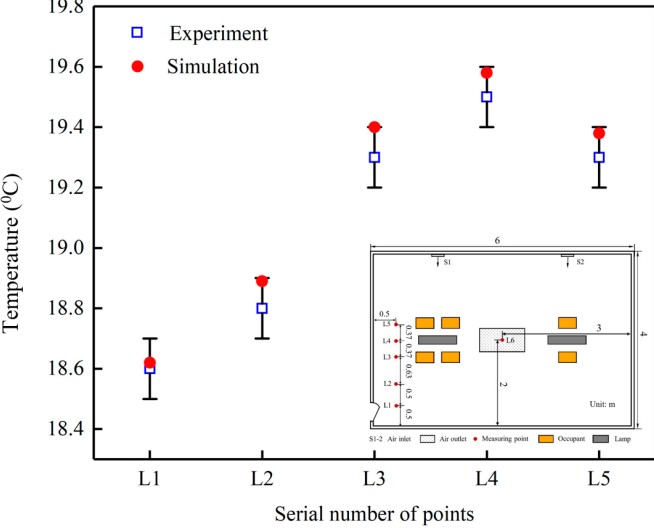

**Figure 6.** Comparisons of experimental and numerical results of temperature.

*3.2. Energy Use Efficiency*

3.2.1. Airflow Distribution

The airflow distribution can be qualitatively analyzed by the velocity cloud atlas and quantitatively calculated by the MAA. The smaller MAA means a more reasonable indoor airflow distribution.

Figure 7 shows velocity distributions in different planes of rooms with different sizes. It can be seen from the $X = 1.5$ m and $X = 4.5$ m plane that the indoor hot-air flowhot-air flow diffusion phenomenon becomes more serious, and the eddy current phenomenon becomes less obvious, with the increase in room size. At the same time, due to the influence of thermal buoyancy, the hot air starts to rise before reaching the occupied zone when the room size reaches 7 m, which indicates that the hot air flow utilization efficiency is reduced.

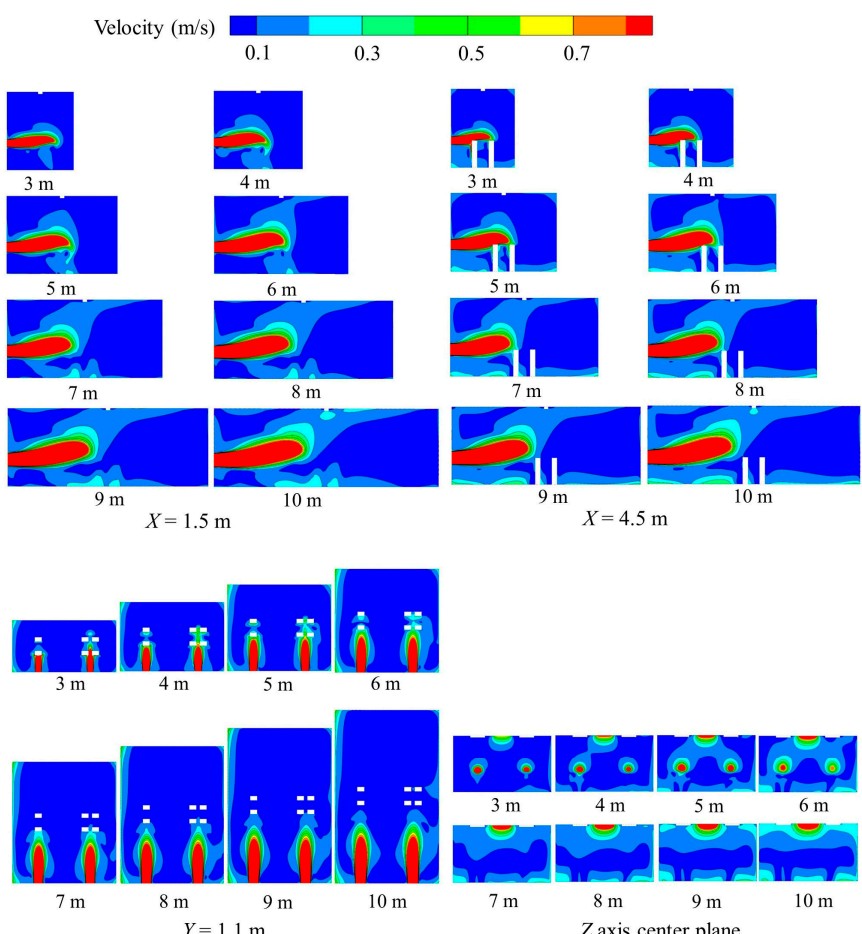

**Figure 7.** Velocity distribution in different room sizes.

In the $Y = 1.1$ m plane, the depth of hot jets in different-sized rooms is the same. However, the proportion of indoor hot-air flow decreases due to different room sizes. Most areas of the room do not exchange heat with the air supply, which leads to the temperature of the area far away from the supply air inlet not being guaranteed.

In the $Z$-axis center plane, when the room size is small, the hot jet can reach the middle of the room. However, with the increase in the size, the air velocity in the middle of the room decreases. This means that hot air begins to diffuse to the non-occupied zone, resulting in a waste of energy.

The air velocity cloud image can only qualitatively analyze the stay of fresh air in the room. The mean age of air was introduced to quantify the residence time of fresh air in the occupied zone. Table 4 reflects the maximum MAA, the average MAA, and the reduction rate of the average MAA in the occupied zone with a height of 0–1.4 m. As shown in

Table 3, the MAA increased from 179.9 s to 248.4 s when the room size increased from 3 m to 10 m, which increased by 38.1%. This means that the indoor ventilation effect of the 3 m room is excellent, and the indoor air quality is high. At the same time, when the room size reaches 7 m, the mean age of air decreases the most, which indicates that 7 m is the maximum applicable size for stratum ventilation.

**Table 4.** Change of MAA with room size.

| Room Size (m) | 3 | 4 | 5 | 6 | 7 | 8 | 9 | 10 |
|---|---|---|---|---|---|---|---|---|
| Maximum MAA (s) | 235.8 | 236.7 | 257.8 | 276.0 | 303.1 | 317.6 | 325.1 | 336.2 |
| Average MAA (s) | 179.9 | 188.9 | 200.9 | 213.9 | 228.8 | 237.6 | 243.8 | 248.4 |
| Reduction rate (%) | | 5.01 | 6.36 | 6.46 | 6.97 | 3.84 | 2.64 | 1.89 |

### 3.2.2. Ventilation Efficiency

Figure 8 shows the change in ventilation efficiency in different room sizes. Ventilation efficiency can indicate the stay of hot-air flow in the occupied zones of the room. The higher the ventilation efficiency, the better the effective utilization of hot-air flow.

As shown in Figure 8, the ventilation efficiency showed a downward trend. The ventilation efficiency decreased from 90.4% to 74.40% when the room width increased from 3 m to 10 m. Compared with the 3 m room, the 10 m room decreased by 17.70%. Referring to the first derivative, the room ventilation efficiency decreases fastest at the room size of 5.7 m.

This is due to the hot-air flow being mainly dominated by its inertia force, forming an eddy current, and staying in the occupied zone when the room size is small. When the size of the room becomes larger, the hot-air flow will spread to the whole room due to the thermal buoyancy and the attraction of the return air outlet. In addition, as the room width becomes larger, the return air temperature becomes higher and higher, the average temperature in the working area becomes lower, and the ventilation efficiency will become smaller.

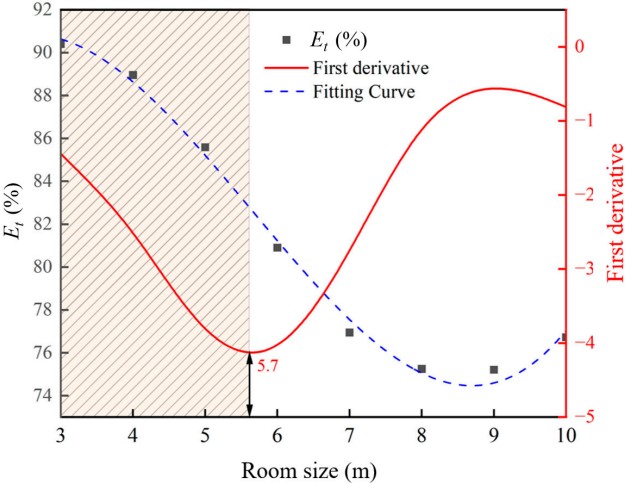

**Figure 8.** Change of ventilation efficiency with room size.

The increasing size will lead to an unreasonable air distribution in the room, and referring to decreased speed, it is advisable to control the width of the room organized by stratum ventilation airflow within 5.7 m as much as possible.

### 3.2.3. Dimensionless Temperature

Figure 9 shows the change of dimensionless temperature in different room sizes. Through dimensionless temperature, the uniformity of room temperature can be reflected

directly, and the regulation of jet on the room environment can be characterized indirectly. With the increase of room size from 3 m to 10 m, the dimensionless temperature gradually decreases. The mean value of dimensionless temperature decreased from 0.96 to 0.85, a decrease of 11.5%. This shows that the increase in room size will lead to the aggravation of thermal stratification. This happens because when the room is too large, the hot-air flow cannot be sent to the back of the room, which weakens the temperature regulation of the area. Moreover, with the increase in room size, the rising range of hot-air flow is obvious, leading to uneven temperature distribution and serious stratification.

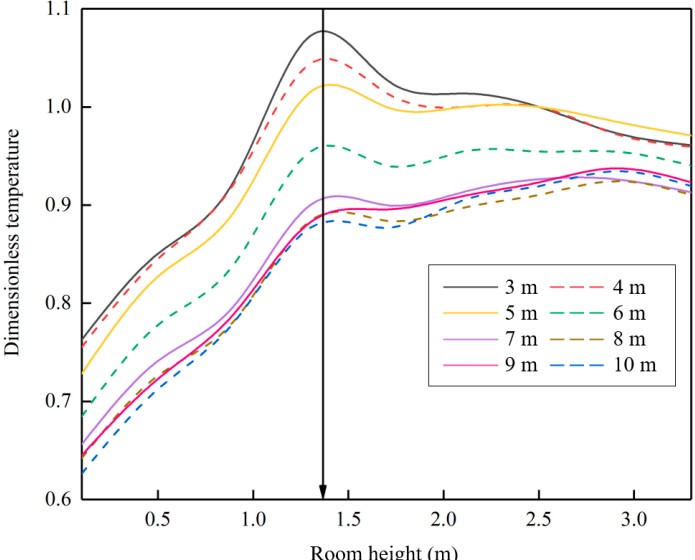

**Figure 9.** Change of dimensionless temperature at different heights with various room sizes.

Taking a height slightly higher than the fresh air inlet as the demarcation point, the dimensionless temperature below 1.3 m has an extensive variation range, while the above variation range is small. The average change rate is 38.6% from 0 m to 1.3 m, and only 4.8% from 3 m to 3.3 m. This is mainly due to the rise of airflow caused by thermal buoyancy, which leads to the lack of regulation of hot-air flow in the lower part of the tuyere, and the temperature being more unstable. In order to achieve a better heating effect by stratum ventilation, it is very important to control the hot air floating effectively.

It can be seen from the figure that the thermal stratification is the lightest when the room width is 3 m, 4 m and 5 m, and moderate when it is 6 m. When the room width reaches 7 m, 8 m and 9 m, the thermal stratification is serious. Therefore, from the thermal stratification, it is advisable to control the width of the room organized by stratum ventilation airflow within 6 m as much as possible.

### 3.3. Thermal Comfort

3.3.1. Effective Draft Temperature

Figure 10 shows the EDT distribution cloud atlas in a different plane of 3–10 m room. Red and blue represent unsatisfactory areas of EDT. By comparing the satisfaction rate of EDT in the occupied zones, it was found that the satisfaction rate of EDT increased by 38.2% with the increase of room width from 3 m to 10 m.

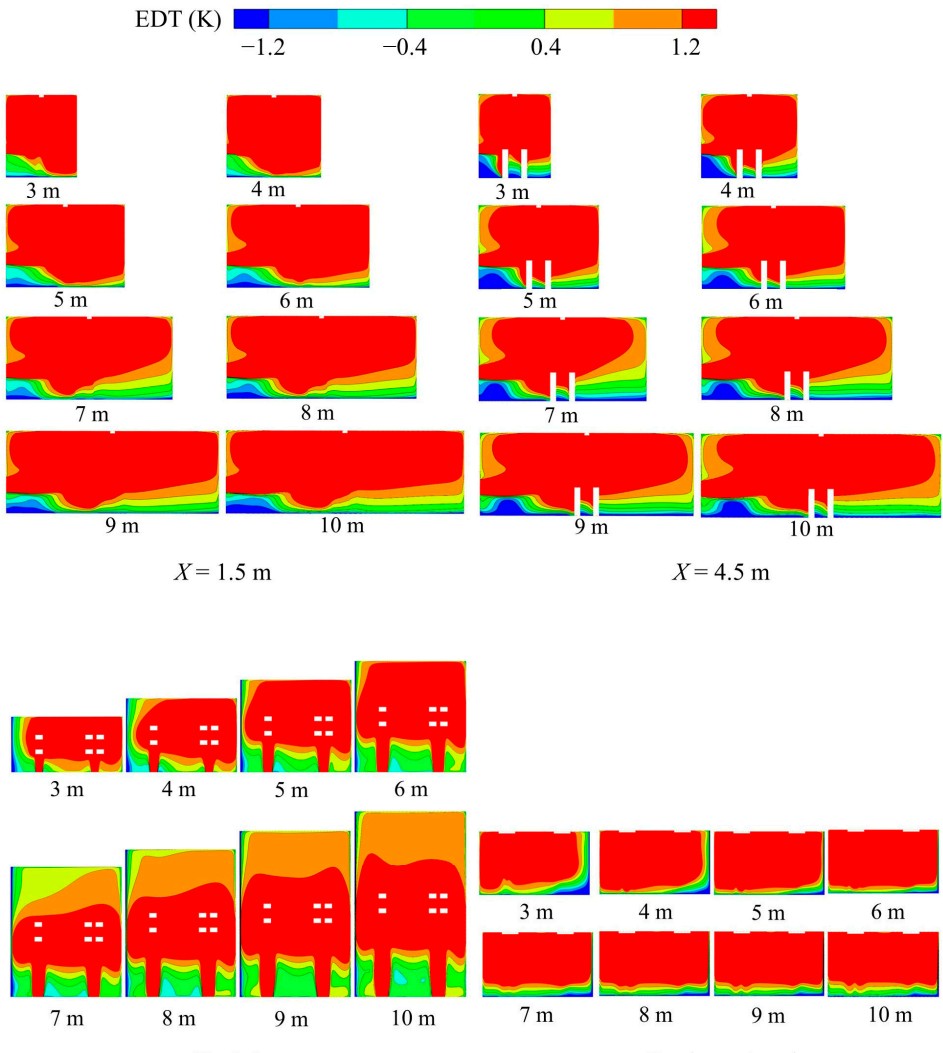

**Figure 10.** EDT distribution in different room size.

With the increase in room size, the phenomenon of airflow diffusion will be obvious. Because EDT is used to describe the uniformity of indoor air temperature and velocity, airflow diffusion will make it more reasonable. At the same time, due to the weakening of the eddy current phenomenon in large-sized rooms, the room temperature gradually decreases, which will also cause a decrease in EDT. After the room size exceeds 7 m, the growth rate decreases, indicating that the maximum applicable size for stratum ventilation is 7 m.

### 3.3.2. ADPI

Figure 11 shows the change of ADPI in the occupied zone under different room sizes. Through the analysis of ADPI, the areas and points that meet both air velocity and draft sensation in different-sized rooms are judged, and the most satisfactory working conditions for the residents in air-conditioned areas can be found.

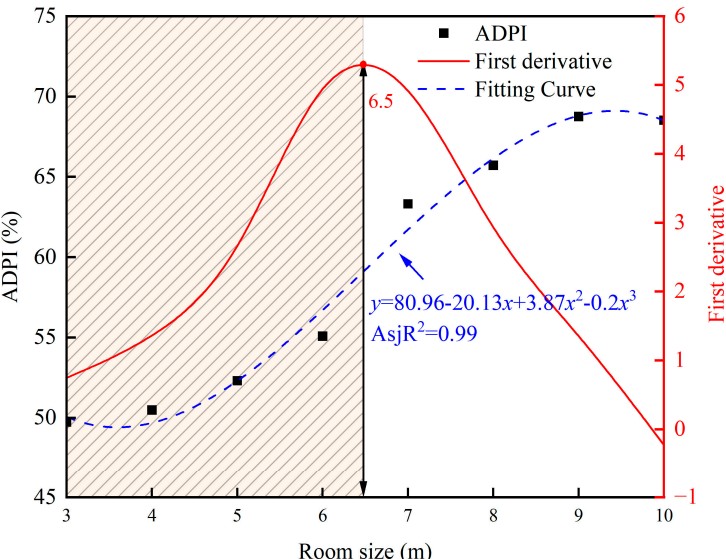

**Figure 11.** ADPI distribution of rooms with different widths.

To further verify the indoor air temperature and draft sensation, as shown in Figure 11, ADPI is selected as an index to evaluate rooms with different sizes. The result is the same as that shown in EDT: the ADPI increases with the increase in room size. When the room size increases from 3 m to 10 m, the ADPI increases from 49.7% to 68.5%. This shows that increasing the size of the room properly will make the room more comfortable. At the same time, when the room width is 6.5 m, the change rate of ADPI is most obvious. By reviewing the relationship between air velocity and temperature comfort level, it is advisable to control the width of the room organized by stratum ventilation airflow within 6.5 m as much as possible.

### 3.3.3. PMV

Figure 12 shows the PMV value along with the change in room size. The PMV index can measure most people's cold and hot feelings in the same environment. The closer the PMV is to 0, the more comfortable the room is.

It can be seen from the figure that 25–75% points of PMV in the room are within the range of −1 to 0, indicating that the whole room meets the comfort zone of PMV. With the increase in room size, PMV decreases gradually, indicating that the risk of room overcooling increases. When the room size is 10 m, the percentage of PMV from −1 to 1 is only 79.9%. Compared with 85.1% of 3 m, the room size of 10 m decreased the PMV by 6.3%. This is due to the smaller room size making the hot-air flow form a vortex in the room, making the heat stay in the occupied zone better. However, with the increase in room size, the hot-air flow will rise due to the buoyancy force, which will reduce the heat in the occupied zone, thus, increasing the cold feeling of the human body. In addition, when the room width is 6 m, the average change rate of PMV reaches the maximum. It shows that PMV is greatly affected by room size. Therefore, to ensure the comfort of indoor persons, the room's width should not exceed 6 m.

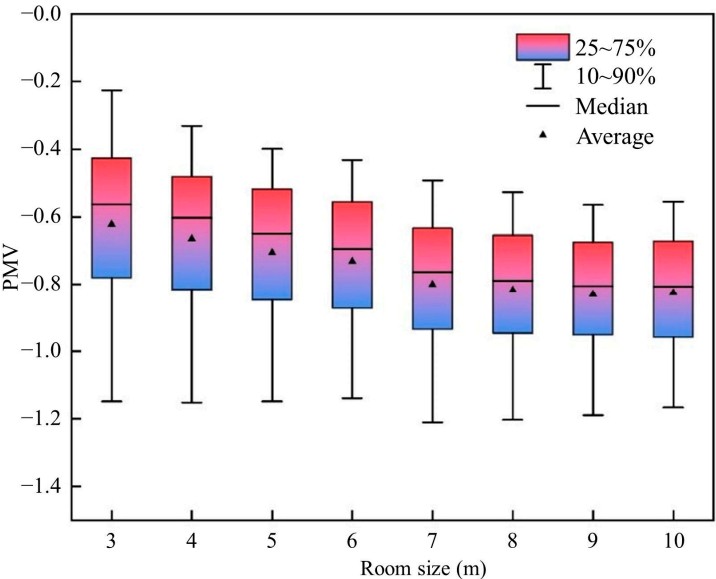

**Figure 12.** PMV distribution in rooms with different widths.

### 3.4. Multi-Criteria AHP-EW Method

3.4.1. AHP-EW Comprehensive Score

To calculate the index scores of different room size dimensions, the data were collected first. The data comes from the the CFD simulation above. The data of each index was selected as follows: the airflow distribution was quantified by the average value of MAA; dimensionless temperature was quantified by the average value of itself; EDT and PMV were quantified by the ratio within the appropriate range; ADPI and $E_t$ were quantified by their values.

Subjective weight was obtained by the AHP method. To make the results professional and accurate, eight experts from universities and engineering fields who are specialized in the ventilation field were invited to score each index. Objective weights were obtained by the EW method. Figure 13a shows the subjective weight, objective weight and comprehensive weight of the six indices.

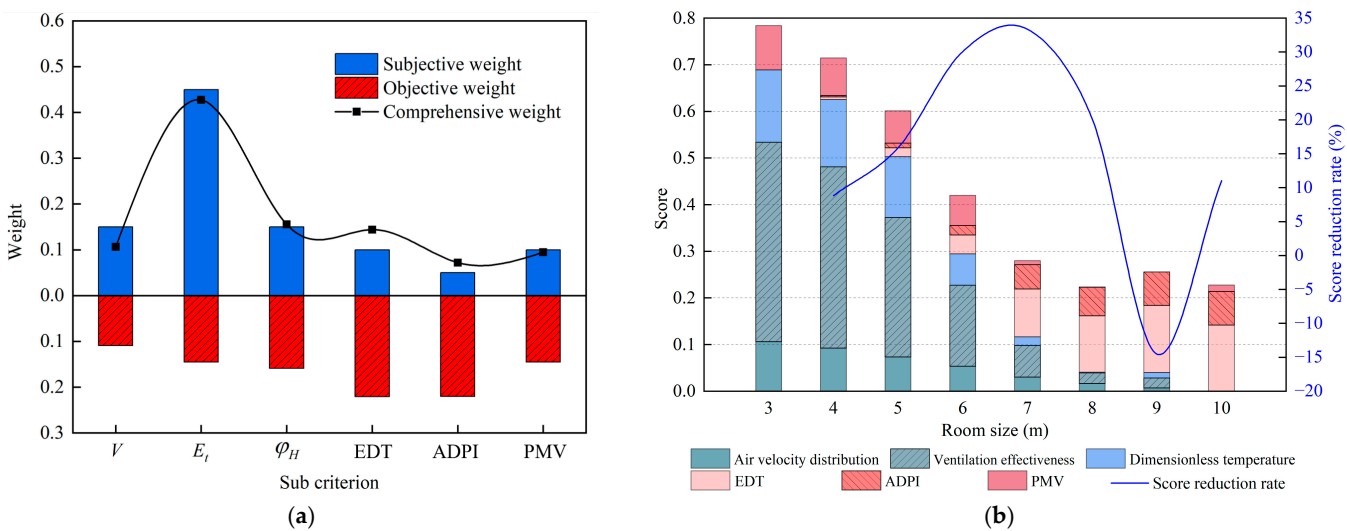

**Figure 13.** (**a**) Weight of the indicators; (**b**) Weight scores of different-sized rooms.

Figure 13b shows the comprehensive weight scores and the six index weight scores of different-sized rooms. The blue part represents the score of energy utilization efficiency; the red part represents thermal comfort score; the line chart represents the comprehensive

score reduction rate. It can be seen from the figure that the proportion of energy utilization efficiency is more significant than that of thermal comfort. This indicated that people always focus on energy consumption when evaluating a room heating system, which is directly related to the current energy shortage situation.

The energy utilization efficiency decreases with the increase in room size, while the thermal comfort reaches the best state in the 9 m room due to the influence of the draft sensation. The comprehensive score decreases with the increase of room size after using the AHP-EW method to analyze the comprehensive weight on each index. The reduction rate of the comprehensive score was analyzed to obtain the applicable ventilation size. The reduction rate is the largest at a room size of 6 m, reaching 30.9%, indicating that the data fluctuates the most in this case. This indicates that a room size exceeding 6 m will cause a decrease in energy utilization efficiency and thermal comfort, which is not conducive to the rational use of stratum ventilation.

### 3.4.2. Determining the Applicable Room Size Dimension

Table 5 lists the maximum applicable room sizes of stratum ventilation with different indices. The values calculated by the multi-criteria AHP-EW model are marked to show the final results more clearly. In order to meet energy utilization efficiency and thermal comfort, the maximum size of the stratum ventilation room should not exceed 5.7 m.

**Table 5.** Determination of applicable room size dimension for stratum ventilation used for heating.

| Evaluation Criteria | Applicable Room Size Dimension (m) |
| --- | --- |
| Airflow distribution | <7 |
| $E_t$ | <5.7 |
| Dimensionless temperature | <6 |
| EDT | <7 |
| ADPI | <6.5 |
| PMV | <6 |
| AHP-EW model | <6 |
| Multi-criteria AHP-EW model | <5.7 |

### 3.5. Sensitivity Analysis

Sensitivity analysis was applied in this study to eliminate the influence of air supply temperature and air supply velocity on the results. Sensitivity analysis compared the applicable room size dimension of stratum ventilation for space heating with an air supply temperature of 26–29 °C and an air supply velocity of 1–2 m/s [54]. When calculating the effect of one parameter, the other parameter remained unchanged [55]. To ensure the accuracy of the results, ventilation efficiency and dimensionless temperature, the two indices with the greatest weight in the AHP-EW method, are used to evaluate the effect of air supply temperature and velocity.

Figure 14a,b show the applicable room size dimensions of stratum ventilation under different air supply temperatures and velocities. Under different air supply temperatures and velocities, the ventilation efficiency and dimensionless temperature change trends are the same, and the maximum change rates appear at 5.7 m and 6 m. It indicates that the temperature and speed of the air supply will not affect the applicable room size dimension of stratum ventilation for space heating.

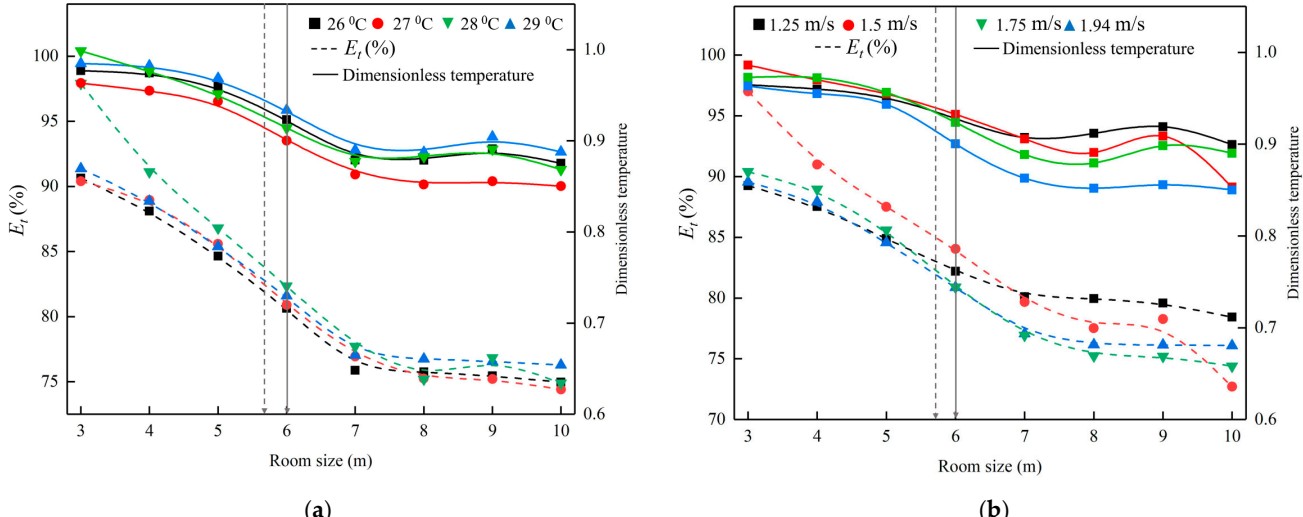

**Figure 14.** (**a**) The applicable room size dimensions under different air supply temperatures; (**b**) The applicable room size dimensions under different air supply velocity.

## 4. Conclusions

In this study, the applicable room size dimension of stratum ventilation for space heating is determined by using a multi-criteria analytic hierarchy process-entropy weight (AHP-EW) model. Energy utilization efficiency and thermal comfort were compared through CFD numerical simulation, and the following conclusions are drawn:

(1) Increasing room size caused increasingly significant hot air floating. An increase of room size by 3.3 times would reduce the energy utilization efficiency by 11.5–37.5%.

(2) In order to ensure the energy efficiency of stratum ventilation, the room size should not exceed 5.7 m.

(3) As the room size increased from 3 m to 10 m, the temperature and velocity uniformity increased by 37.8–38.2%, but the risk of overcooling also increased by 32.6%.

(4) Considering thermal comfort, the maximum applicable sizes for stratum ventilation were 6 m.

(5) Using the multi-indicator AHP-EW method for the combined evaluation of energy use efficiency and thermal comfort found that the score decreases by 9% with room size increasing from 3 m to 10 m. By coupling single criteria, the maximum applicable size of stratum ventilation of 5.7 m is obtained.

To sum up, stratum ventilation is suitable for general offices with a width of 3–4 m but not exceeding 5.7 m. However, for larger office heating in winter, the performance of this ventilation system still needs to be improved.

**Author Contributions:** Conceptualization, Y.M.; methodology, Y.M. and H.X.; software, Y.M., X.Z. and M.W.; validation, X.Z.; formal analysis, Y.M. and H.X.; resources, M.W.; data curation, F.H. and M.W.; writing—original draft preparation, Y.M.; writing—review and editing, Y.M.; project administration, Y.M.; funding acquisition, Y.M. All authors have read and agreed to the published version of the manuscript.

**Funding:** This work was supported by the Zhejiang Provincial Natural Science Foundation (No. LGF21E080007).

**Institutional Review Board Statement:** Not applicable.

**Informed Consent Statement:** Not applicable.

**Data Availability Statement:** The data presented in this study are available on request from the corresponding authors.

**Conflicts of Interest:** The authors declare no conflict of interest.

**Abbreviations**

| | |
|---|---|
| $S_E$ | fluid energy source |
| $E_t$ | ventilation effectiveness (%) |
| EDT | effective draft temperature (K) |
| ADPI | air distribution performance index (%) |
| PMV | predicted mean vote |
| MAA | the mean age of air (s) |
| ACH | air changes per hour |
| $\rho$ | density of the fluid (kg/m$^3$) |
| $t$ | time (s) |
| $U$ | fluid flow velocity (m/s) |
| $\tau$ | stress tensor |
| $T$ | Temperature (°C) |
| $\delta$ | identity matrix |
| $P$ | thermodynamic compressive strength (Pa) |
| $S_M$ | fluid momentum source |
| $h$ | fluid static enthalpy (J/kg) |
| $\lambda$ | fluid thermal conductivity (w/(m·k)) |
| $P$ | experimental measuring points number |
| $\tau_p$ | the local mean age of air (s) |
| $C_{t-t0}$ | the initial tracer gas concentration at time $t$-$t_0$ |
| $T_s$ | supply air temperature (°C) |
| $T_e$ | return air temperature (°C) |
| $T_o$ | average temperature in the occupied zone (°C) |
| $T_x$ | temperature of a certain point in space (°C) |
| $T_r$ | average indoor temperature (°C) |
| $x_i$ | experimental result (°C) |
| $y_i$ | numerical simulation result (°C) |
| $N_s$ | total number of effective blowing temperature and air velocity in the comfortable range |
| $x_{ij}{}'$ | The standardized score of the $j_{th}$ room size in the $i_{th}$ index |
| $x_{ij}$ | initial score of the $j_{th}$ room size in the $i_{th}$ index |
| $k$ | coefficients |
| $q$ | coefficients |
| $\alpha_i$ | AHP weight of $i_{th}$ index |
| $W_i$ | EW weight of $i_{th}$ index |
| $\beta_i$ | comprehensive weight of $i_{th}$ index |
| $S_j$ | comprehensive weight score for the $j_{th}$ size room |
| $E$ | relative error (%) |
| $\Delta$ | absolute error (°C) |
| $L$ | True value (°C) |
| $\varphi_H$ | dimensionless temperature |
| MAPE | mean absolute percentage error |
| $\nu_x$ | air velocity at a certain point in space (m/s) |
| $N$ | total number of occupied zone |
| IPCC | Intergovernmental Panel on Climate Change |
| GHG | greenhouse gas |
| COVID-19 | Corona Virus Disease 2019 |
| AHP | analytic hierarchy process |
| EW | entropy weight |
| CFD | Computational Fluid Dynamics |

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
