# Peer review of "Study on the Applicable Room Size Dimension of Stratum Ventilation for Heating Based on Multi-Criteria Analytic Hierarchy Process-Entropy Weight Model"

_buildings, doi:10.3390/buildings13020381_

Round 1
Reviewer 1 Report
Authors insist that the paper investigates the relevant room dimensions of stratum ventilation for space heating by employing a multi-criteria analytic hierarchy process-entropy weight (AHP-EW) prototype through CFD simulation and tests.
I checked the methodology and results. The point and structure are a good one to determine the relevant room dimensions of stratum ventilation for space heating. Especially, this paper utilized various methods such as Computational fluid dynamics (CFD) simulation, Simulated cases, Ventilation performance evaluation method - Single criteria & Multi-criteria (AHP-EW 295 model),
However, this paper looks like a design report for optimal building that satisfies the comfort. There is no clear explanation what stratum ventilation and its effectiveness in air conditioning are. And as a research paper, it requires its own methodology, not other research. So, your own methodology with title in section 2 should described.
I recommend to modify the papers. My judgments come from as following:
1) As the emphasis of the paper is on this study, more introduction to the ALT is required to show the innovation.
2) Your own methodology with title in section 2 should described.
3) Conclusion also is not concise. It is required to distinctively show your research results with using some bullet.
Reviewer 2 Report
Please see the attached file.

Reviewer 3 Report
It is an interesting application of multi-criteria analytic hierarchy process-entropy weight (AHP-EW) method to determine the room size dimension of stratum ventilation for space heating. However, there are some references missed which can be found easily in the literature related to stratum ventilation, for instance: Stratum ventilation — a low-carbon way to thermal comfort and indoor air quality, Zhang Lin, International Journal of Low-Carbon Technologies 2017, 12, 323–329 Therefore I suggest adding references missed to enhance the validity of such contribution.
Reviewer 4 Report
The work is important from the aspect of creating favorable conditions for living and working indoors and has great potential and can be accepted after the following minor corrections:
1. Perform technical processing of formulas in the text! (number 3 cannot be followed by number 13)
2. It is necessary to arrange Table 2 - Na letter (Numeric parameters) are not visible
3. It is necessary to arrange Table 4 - Why is the "Multi-criteria AHP-EW model" colored
4. On what basis are the criteria defined?
5. Lines 121-123 are listed criteria that are not described!
6. Line 176 "The velocity of the air flow at the outlet in the model is assumed to be uniform" (Explain on the basis of what?),
7. Line 202: On what basis was the point L6 in the center of the room chosen as the point of comparison! Why is it not marked in the pictures, like the previous 5 points?
8. It was not explained on the basis of which the individual criteria were defined?
9. Line 317-318: Do the results need to be more precise or more convenient to process?
Round 2
Reviewer 1 Report
The raised issues are answered. However, the modified portion is a little wordy. You need to make more concise.
Reviewer 2 Report
The authors implemented the reviewer's suggestions.
Maybe Section 6 can be eliminated from the paper.
